# A Market Framework for Eliciting Private Data

**Bo Waggoner**
Harvard SEAS
bwaggoner@fas.harvard.edu

**Rafael Frongillo**
University of Colorado
raf@colorado.edu

**Jacob Abernethy**
University of Michigan
jabernet@umich.edu

## Abstract

We propose a mechanism for purchasing information from a sequence of participants. The participants may simply hold data points they wish to sell, or may have more sophisticated information; either way, they are incentivized to participate as long as they believe their data points are representative or their information will improve the mechanism's future prediction on a test set. The mechanism, which draws on the principles of prediction markets, has a bounded budget and minimizes generalization error for Bregman divergence loss functions. We then show how to modify this mechanism to preserve the privacy of participants' information: At any given time, the current prices and predictions of the mechanism reveal almost no information about any one participant, yet in total over all participants, information is accurately aggregated.

## 1 Introduction

A firm that relies on the ability to make difficult predictions can gain a lot from a large collection of data. The goal is often to estimate values $y \in \mathcal{Y}$ given observations $x \in \mathcal{X}$ according to an appropriate class of *hypotheses* $\mathcal{F}$ describing the relationship between $x$ and $y$ (for example, $y = a \cdot x + b$ for linear regression). In classic statistical learning theory, the goal is formalized as attempting to approximately solve

$$\min_{f \in \mathcal{F}} \; \mathbb{E}_{x,y} \; \text{Loss}(f; (x,y)) \tag{1}$$

where $\text{Loss}(\cdot)$ is an appropriate inutility function and $(x,y)$ is drawn from an unknown distribution.

In the present paper we are concerned with the case in which the data are not drawn or held by a central authority but are instead *inherently distributed*. By this we mean that the data is (disjointly) partitioned across a set of agents, with agent $i$ privately possessing some portion of the dataset $S_i$, and agents have no obvious incentive to reveal this data to the firm seeking it. The vast swaths of data available in our personal email accounts could provide massive benefits to a range of companies, for example, but users are typically loathe to provide account credentials, even when asked politely.

We will be concerned with the design of financial mechanisms that provide a community of agents, each holding a private set of data, an incentive to contribute to the solution of a large learning or prediction task. Here we use the term 'mechanism' to mean an algorithmic interface that can receive and answer queries, as well as engage in monetary exchange (deposits and payouts). Our aim will be to design such a mechanism that satisfies the following three properties:

1. The mechanism is *efficient* in that it approaches a solution to (1) as the amount of data and participation grows while spending a constant, fixed total budget.
2. The mechanism is *incentive-compatible* in the sense that agents are rewarded when their contributions provide marginal value in terms of improved hypotheses, and are not rewarded for bad or misleading information.
3. The mechanism provides reasonable *privacy guarantees*, so that no agent $j$ (or outside observer) can manipulate the mechanism in order to infer the contributions of agent $i \neq j$.

Ultimately we would like our mechanism to approach the performance of a learning algorithm that had direct access to all the data, while only spending a constant budget to acquire data and improve predictions and while protecting participants' privacy.

Our construction relies on the recent surge in literature on *prediction markets* [13, 14, 19, 20], popular for some time in the field of economics and recently studied in great detail in computer science [8, 16, 6, 15, 18, 1]. A prediction market is a mechanism designed for the purpose of information aggregation, particularly when there is some underlying future event about which many members of the population may have private and useful information. For instance, it may elicit predictions about which team will win an upcoming sporting event, or which candidate will win an election. These predictions are eventually scored on the actual outcome of the event.

Applying these prediction market techniques allows participants to essentially "trade in a market" based on their data. (This approach is similar to prior work on crowdsourcing contests [3].) Members of the population have private information, just as with prediction markets — in this case, data points or beliefs — and the goal is to incentivize them to reveal and aggregate that information into a final hypothesis or prediction. Their final profits are tied to the outcome of a test set of data, with each participant being paid in accordance with how much their information improved the performance on the test set. Our techniques depart from the framework of [3] in two significant aspects: (a) we focus on the particular problem of data aggregation, and most of our results take advantage of kernel methods; and (b) our mechanisms are the first to combine differential privacy guarantees with data aggregation in a prediction-market framework.

This framework will provide efficiency and truthfulness. We will also show how to achieve privacy in many scenarios. We will give mechanisms where the prices and predictions published satisfy $(\epsilon, \delta)$-*differential privacy* [10] with respect to each participant's data. The mechanism's output can still give reasonable predictions while no observer can infer much about any participant's input data.

## 2 Mechanisms for Eliciting and Aggregating Data

We now give a broad description of the mechanism we will study. In brief, we imagine a central authority (the mechanism, or market) maintaining a hypothesis $f^t$ representing the current aggregation of all the contributions made thus far. A new (or returning) participant may query $f^t$ at no cost, perhaps evaluating the quality of the predictions on a privately-held dataset, and can then propose an update $df^{t+1}$ to $f^t$ that possibly requires an investment (a "bet"). Bets are evaluated at the close of the market when a true data sample is generated (analogous to a test set), and payouts are distributed according to the quality of the updates.

After describing this initial framework as Mechanism 1, which is based loosely on the setting of [3], we turn our attention to the special case in which our hypotheses must lie in a Reproducing Kernel Hilbert Space (RKHS) [17] for a given kernel $k(\cdot, \cdot)$. This kernel-based "nonparametric mechanism" is particularly well-suited for the problem of *data aggregation*, as the betting space of the participants consists essentially of updates of the form $df^t = \alpha_t k(z_t, \cdot)$, where $z_t$ is the data object offered by the participant and $\alpha_t \in \mathbb{R}$ is the "magnitude" of the bet.

A drawback of Mechanism 1 is the lack of privacy guarantees associated with the betting protocol: utilizing one's data to make bets or investments in the mechanism can lead to a loss of privacy for the owner of that data. When a participant submits a bet of the form $df^t = \alpha_t k(z_t, \cdot)$, where $z_t$ could contain sensitive personal information, another participant may be able to infer $z_t$ by querying the mechanism. One of the primary contributions of the present work, detailed in Section 3, is a technique to allow for productive participation in the mechanism while maintaining a guarantee on the privacy of the data submitted.

### 2.1 The General Template

There is a space of examples $\mathcal{X} \times \mathcal{Y}$, where $x \in \mathcal{X}$ are features and $y \in \mathcal{Y}$ are labels. The mechanism designer chooses a function space $\mathcal{F}$ consisting of $f : \mathcal{X} \times \mathcal{Y} \to \mathbb{R}$, and assumed to have Hilbert space structure; one may view $\mathcal{F}$ as either the hypothesis class or the associated *loss class*, that is where $f_h(x, y)$ measures the loss/performance of hypothesis $h$ on observation $x$ and label $y$. In each case we will refer to $f \in \mathcal{F}$ as a hypothesis, eliding the distinction between $f_h$ and $h$.

The pricing scheme of the mechanism relies on a convex *cost function* $C_x(\cdot) : \mathcal{F} \to \mathbb{R}$ which is parameterized by elements $x \in \mathcal{X}$ but whose domain is the set of hypotheses $\mathcal{F}$. The cost function is publicly available and determined in advance. The interaction with the mechanism is a sequential process of *querying* and *betting*. On round $t-1$ the mechanism publishes a hypothesis $f^{t-1}$, the "state" of the market, which participants may query. Each participant arrives sequentially, and on round $t$ a participant may place a "bet" $df^t \in \mathcal{F}$, also called a "trade" or "update", modifying the hypothesis $f^{t-1} \to f^t = f^{t-1} + df^t$. Finally participation ends and the mechanism samples (or reveals) a test example[1] $(x, y)$ from the underlying distribution and pays (or charges) each participant according to the relative performance of their marginal contributions. Precisely, the total reward for participant $t$'s bet $df^t$ is the value $df^t(x,y)$ minus the cost $C_x(f^t) - C_x(f^{t-1})$.

---

**Mechanism 1:** The Market Template

---

MARKET announces $f^0 \in \mathcal{F}$
**for** $t = 1, 2, \ldots, T$ **do**
  PARTICIPANT may query functions $\nabla_f C_x(f^{t-1})$ and $f^{t-1}(x,y)$ for examples $(x,y)$
  PARTICIPANT $t$ may submit a *bet* $df^t \in \mathcal{F}$ to MARKET
  MARKET updates state $f^t = f^{t-1} + df^t$
MARKET observes a true sample $(x,y)$
**for** $t = 1, 2, \ldots, T$ **do**
  PARTICIPANT $t$ receives payment $df^t(x,y) + C_x(f^{t-1}) - C_x(f^t)$

---

The design of cost-function prediction markets has been an area of active research over the past several years, starting with [8] and many further refinements and generalizations [1, 6, 15]. The general idea is that the mechanism can efficiently provide price quotes via a function $C(\cdot)$ which acts as a potential on the space of outstandings shares; see [1] for a thorough review. In the present work we have added an additional twist which is that the function $C_x(\cdot)$ is given an additional parameterization of the observation $x$. We will not dive too deeply into the theoretical aspects of this generalization, but this is a straightforward extension of existing theory.

**Key special case: exponential family mechanism.** For those more familiar with statistics and machine learning, there is a natural and canonical family of problems that can be cast within the general framework of Mechanism 1, which we will call the *exponential family prediction mechanism* following [2]. Assume that $\mathcal{F}$ can be parameterized as $\mathcal{F} = \{f_\theta : \theta \in \mathbb{R}^d\}$, that we are given a sufficient statistics summary function $\phi : \mathcal{X} \times \mathcal{Y} \to \mathbb{R}^d$, and that function evaluation is given by $f_\theta(x,y) = \langle \theta, \phi(x,y) \rangle$. We let $C_x(f) := \log \int_{\mathcal{Y}} \exp(f(x,y)) dy$ so that $C_x(f_\theta) = \log \int_{\mathcal{Y}} \exp(\langle \theta, \phi(x,y) \rangle) dy$. In other words, we have chosen our mechanism to encode a particular exponential family model, with $C_x(\cdot)$ chosen as the conditional log partition function over the distribution on $y$ given $x$. If the market has settled on a function $f_\theta$, then one may interpret that as the aggregate market belief on the distribution of $\mathcal{X} \times \mathcal{Y}$ is

$$p_\theta(x,y) = \exp(\langle \theta, \phi(x,y) \rangle - A(\theta)) \qquad \text{where} \qquad A(\theta) = \log \int_{\mathcal{X} \times \mathcal{Y}} \exp(\langle \theta, \phi(x,y) \rangle) \, dx \, dy.$$

How may we view this as a "market aggregate" belief? Notice that if a trader observes the market state of $f_\theta$ and she is considering a bet of the form $df = f_\theta - f_{\theta'}$, the eventual profit will be

$$f_{\theta'}(x,y) - f_\theta(x,y) + C_x(f_\theta) - C_x(f_{\theta'}) = \log \frac{p_{\theta'}(y|x)}{p_\theta(y|x)}.$$

I.e., the profit is precisely the conditional log likelihood ratio of the update $\theta \to \theta'$.

**Example: Logistic regression.** Let $\mathcal{X} = \mathbb{R}^k$, $\mathcal{Y} = \{-1, 1\}$, and take $\mathcal{F}$ to be the set of functions $f_\theta(x,y) = y \cdot (\theta^\top x)$ for $\theta \in \mathbb{R}^k$. Then by our construction, $C_x(f) = \log(\exp(f(x,1)) + \exp(f(x,-1))) = \log(\exp(\theta^\top x) + \exp(-\theta^\top x))$, and we let $f^0 = f_0 \equiv 0$. The payoff of a participant placing a bet which moves the market state to $f^1 = f_\theta$, upon outcome $(x,y)$, is:

$$f_\theta(x,y) + C_x(f_0) - C_x(f_\theta) = y\theta^\top x + \log(2) - \log(\exp(\theta^\top x) + \exp(-\theta^\top x))$$

$$= \log(2) - \log(1 + \exp(-2y\theta^\top x)) ,$$

which is simply negative logistic loss of the parameter choice $2\theta$. A participant wishing to maximize profit under a belief distribution $p(x, y)$ should therefore choose $\theta$ via logistic regression,

$$\theta^* = \arg \min_\theta \mathbb{E}_{(x,y)\sim p} \left[ \log(1 - \exp(2y\theta^\top x)) \right] . \tag{2}$$

## 2.2 Properties of the Market

We next describe two nice properties of Mechanism 1: *incentive-compatibility* and *bounded budget*. Recall that, for the exponential family markets discussed above, a trader moving the market hypothesis from $f^{t-1}$ to $f^t$ was compensated according to the conditional log-likelihood ratio of $f^{t-1}$ and $f^t$ on the test data point. The implication is that traders are incentivized to minimize a KL divergence between the market's estimate of the distribution and the true underlying distribution. We refer to this property as incentive-compatibility because traders' interests are aligned with the mechanism designer's. This property indeed holds generally for Mechanism 1, where the KL divergence is replaced with a general *Bregman divergence* corresponding to the Fenchel conjugate of $C_x(\cdot)$; see Proposition 1 in the appendix for details.

Given that the mechanism must make a sequence of (possibly negative) payments to traders, a natural question is whether there is the potential for large downside for the mechanism in terms of total payment (budget). In the context of the exponential family mechanism, this question is easy to answer: after a sequence of bets moving the market state parameter $\theta_0 \to \theta_1 \to \ldots \to \theta_{\text{final}}$, the total loss to the mechanism corresponds to the total payouts made to traders,

$$\sum_i f_{\theta_{i+1}}(x, y) - f_{\theta_i}(x, y) + C_x(f_{\theta_i}) - C_x(f_{\theta_{i+1}}) = \log \frac{p_{\theta_{\text{final}}}(y|x)}{p_{\theta_0}(y|x)};$$

that is, the worst-case loss is exactly the worst-case conditional log-likelihood ratio. In the context of logistic regression this quantity can always be guaranteed to be no more than $\log 2$ as long as the initial parameter is set to $\theta = 0$. For Mechanism 1 more generally, one has tight bounds on the worst-case loss following from such results from prediction markets [1, 8], and we give a more detailed statement in Proposition 2 in the appendix.

**Price sensitivity parameter $\lambda_C$.** In choosing the cost function family $C = \{C_x : x \in \mathcal{X}\}$, an important consideration is the "scale" of each $C_x$, or how quickly changes in the market hypothesis $f^t$ translate to changes in the "instantaneous prices" $\nabla C_x(f^t)$ (which give the marginal cost for an infinitesimal bet $df^{t+1}$). Formally, this is captured by the *price sensitivity* $\lambda_C$, defined as the upper bound on the operator norm (with respect to the $L_1$ norm) of the Hessian of the cost function $C_x$ (over all $x$). A choice of small $\lambda_C$ translates to a small worst-case budget required by the mechanism. However, it means that the market prices are sensitive in that the same update $df^t$ changes the prices much more quickly. When we consider protecting the privacy of trader updates in Section 3, we will see that privacy imposes restrictions on the price sensitivity.

## 2.3 A Nonparametric Mechanism via Kernel Methods

The framework we have discussed thus far has involved a general function space $\mathcal{F}$ as the "state" of the mechanism, and the contributions by participants are in the form of modifications to these functions. One of the downsides of this generic template is that participants may not be able to reason about $\mathcal{F}$, and they may have information about the optimal $f$ only through their own privately-held dataset $S \subset \mathcal{X} \times \mathcal{Y}$. A more specific class of functions would be those parameterized by actual data. This brings us to a well-studied type of non-parametric hypothesis class, namely the reproducing kernel Hilbert space (RKHS). We can design a market based on an RKHS, which we will refer to as a *kernel market*, that brings together a number of ideas including recent work of [21] as well as kernel exponential families [4].

We have a positive semidefinite kernel $k : \mathcal{Z} \times \mathcal{Z} \to \mathbb{R}$ and associated reproducing kernel Hilbert space $\mathcal{F}$, with basis $\{f_z(\cdot) = k(z, \cdot) : z \in \mathcal{Z}\}$. The reproducing property is that for all $f \in \mathcal{F}$, $\langle f, k(z, \cdot) \rangle = f(z)$. Now each hypothesis $f \in \mathcal{F}$ can be expressed as $f(\cdot) = \sum_s \alpha_s k(z_s, \cdot)$ for some collection of points $\{(\alpha_s, z_s)\}$.

The kernel approach has several nice properties. One is a natural extension of the exponential family mechanism using an RKHS as a building block of the class of exponential family distributions [4]. A

key assumption in the exponential family mechanism is that evaluating $f$ can be viewed as an inner product in some feature space; this is precisely what one has given a kernel framework. Specifically, assume we have some PSD kernel $k : \mathcal{X} \times \mathcal{X} \to \mathbb{R}$, where $\mathcal{Y} = \{-1, 1\}$. Then we can define the associated *classification kernel* $\hat{k} : (\mathcal{X} \times \mathcal{Y}) \times (\mathcal{X} \times \mathcal{Y}) \to \mathbb{R}$ according to $\hat{k}((x, y), (x', y')) := yy'k(x, x')$. Under certain conditions [4], we again can take $C_x(f) = \log \int_{\mathcal{Y}} \exp(f(x, y)) dy$, and for any $f$ in the RKHS associated to $\hat{k}$, we have an associated distribution of the form $p_f(x, y) \propto \exp(f(x, y))$. And again, a participant updating the market from $f^{t-1}$ to $f^t$ is rewarded by the conditional log-likelihood ratio of $f^{t-1}$ and $f^t$ on the test data.

The second nice property mirrors one of standard kernel learning methods, namely that under certain conditions one need only search the subset of the RKHS spanned by the basis $\{k((x_i, y_i), \cdot) : (x_i, y_k) \in S\}$, where $S$ is the set of available data; this is a direct result of the *Representer Theorem* [17]. In the context of the kernel market, this suggests that participants need only interact with the mechanism by pushing updates that lie in the span of their own data. In other words, we only need to consider updates of the form $df = \alpha k((x, y), \cdot)$. This naturally suggests the idea of directly purchasing data points from traders.

**Buying Data Points.** So far, we have supposed that a participant knows what trade $df^t$ she prefers to make. But what if she simply has a data point $(x, y)$ drawn from the underlying distribution? We would like to give this trader a "simple" trading interface in which she can sell her data to the mechanism without having to reason about the correct $df^t$ for this data point.

Our proposal is to mimic the behavior of natural learning algorithms, such as stochastic gradient descent, when presented with $(x, y)$. The market can offer the trader the purchase bundle corresponding to the update of the learning algorithm on this data point. In principle, this approach can be used with any online learning algorithm. In particular, stochastic gradient descent gives a clean update rule, which we now describe. The expected profit (which is the negative of expected loss) for trade $df^t$ is $\mathbb{E}_x \left[ C_x(f^{t-1} + df^t) - C_x(f^{t-1}) - \mathbb{E}_{y|x}[df^t(x, y)] \right]$. Given a draw $(x, y)$, the loss function on which to take a gradient step is $- \left( C_x(f^{t-1} + df^t) - C_x(f^{t-1}) - df^t(x, y) \right)$, whose gradient is $-\nabla_{f^{t-1}} C_x + \delta_{x,y}$ (where $\delta_{x,y}$ is the indicator on data point $x, y$). This suggests that the market offer the participant the trade $df^t = \epsilon \left( \nabla_{f^{t-1}} C_x - \delta_{x,y} \right)$, where $\epsilon$ can be chosen arbitrarily as a "learning rate". This can be interpreted as buying a unit of shares in the participant's data point $(x, y)$, then "hedging" by selling a small amount of all other shares in proportion to their current prices (recall that the current prices are given by $\nabla_{f^t} C_x$).

In the kernel setting, the choice of stochastic gradient descent may be somewhat problematic, because it can result in non-sparse share purchases. It may instead be desirable to use algorithms that guarantee sparse updates—a modern discussion of such approaches can be found in [22, 23].

Given this framework, participants with access to a private set of samples from the true underlying distribution can simply opt for this "standard bundle" corresponding to their data point, which is precisely a stochastic gradient descent update. With a small enough learning rate, and assuming that the data point is truly independent of the current hypothesis (i.e. $(x, y)$ has not been previously incorporated), the trade is guaranteed to make at least some positive profit in expectation. More sophisticated alternative strategies are also possible of course, but even the proposed simple bet type has earning potential.

## 3 Protecting Participants' Privacy

We now extend the mechanism to protect privacy of the participants: An adversary observing the hypotheses and prices of the mechanism, and even controlling the trades of other participants, should not be able to infer too much about any one trader's update $df^t$. This is especially relevant when participants sell data to the mechanism and this data can be sensitive, *e.g.* medical data.

Here, privacy is formalized by $(\epsilon, \delta)$-differential privacy, to be defined shortly. One intuitive characterization is that, for any prior distribution some adversary has about a trader's data, the adversary's posterior belief after observing the mechanism would be approximately the same even if the trader did not participate at all. The idea is that, rather than posting the exact prices and trades made in the market, we will publish noisy versions, with the random noise giving the above guarantee.

A naive approach would be to add independent noise to each participant's trade. However, this would require a prohibitively-large amount of noise; the final market hypothesis would be determined by the random noise just as much as by the data and trades. The central challenge is to add carefully correlated noise that is large enough to hide the effects of any one participant's data point, but not so large that the prices (equivalently, hypothesis) become meaningless. We show this is possible by adjusting the "price sensitivity" $\lambda_C$ of the mechanism, a measure of how fast prices change in response to trades defined in 2.2. It will turn out to suffice to set the price sensitivity to be $O(1/\text{polylog } T)$ when there are $T$ participants. This can roughly be interpreted as saying that any one participant does not move the market price noticeably (so their privacy is protected), but just $O(\text{polylog } T)$ traders together can move the prices completely.

We now formally define differential privacy and discuss two useful tools at our disposal.

## 3.1 Differential Privacy and Tools

Differential privacy in our context is defined as follows. Consider a randomized function $M$ operating on inputs of the form $\vec{f} = (df^1, \ldots, df^T)$ and having outputs of the form $s$. Then $M$ is $(\epsilon, \delta)$-*differentially private* if, for any coordinate $t$ of the vector, any two distinct $df_1^t, df_2^t$, and any (measurable) set of outputs $S$, we have $\Pr[M(f^{-t}, df_1^t) \in S)] \leq e^\epsilon \Pr[M(f^{-t}, df_2^t) \in S] + \delta$. The notation $f^{-t}$ means the vector $\vec{f}$ with the $t$th entry removed.

Intuitively, $M$ is private if modifying the $t$th entry in the vector to a different entry does not change the distribution on outputs too much. In our case, the data to be protected will be the trade $df^t$ of each participant $t$, and the space of outputs will be the entire sequence of prices/predictions published by the mechanism.

To preserve privacy, each trade must have a bounded size (*e.g.* consist only of one data point). To enforce this, we define the following parameter chosen by the mechanism designer:

$$\Delta = \max_{\text{allowed } df} \sqrt{\langle df, df \rangle}, \tag{3}$$

where the maximum is over all trades $df$ allowed by the mechanism. That is, $\Delta$ is a scalar capturing the maximum allowed size of any one trade. For instance, if all trades are restricted to be of the form $df = \alpha k(z, \cdot)$, then we would have $\Delta = \max_{\alpha, z} \alpha \sqrt{k(z, z)}$.

We next describe the two tools we require.

**Tool 1: Private functions via Gaussian processes.** Given a current market state $f^t = f^0 + df^1 + \cdots + df^t$, where $f^t$ lies in a RKHS, we construct a "private" version $\hat{f}^t$ such that queries to $\hat{f}^t$ are "accurate" — close to the outputs of $f^t$ — but also private with respect to each $df^j$. In fact, it will become convenient to privately output partial sums of trades, so we wish to output a $\hat{f}_{t_1:t_2}$ that is private and approximates $f_{t_1:t_2} = \sum_{j=t_1}^{t_2} df^j$. This is accomplished by the following construction due to [11].

**Theorem 1** ([11], Corollary 9). *Let $G$ be the sample path of a Gaussian process with mean zero and whose covariance is given by the kernel function $k$.[2] Then*

$$\hat{f}_{t_1:t_2} = f_{t_1:t_2} + \Delta \frac{\sqrt{2 \ln(2/\delta)}}{\epsilon} G . \tag{4}$$

*is $(\epsilon, \delta)$-differentially private with respect to each $df^j$ for $j \in \{t_1, \ldots, t_2\}$.*

In general, $\hat{f}_{t_1:t_2}$ may be an infinite-dimensional object and thus impossible to finitely represent. In this case, the theorem implies that releasing the results of any number of queries $\hat{f}_{t_1:t_2}(z)$ is differentially private. (Of course, the more queries that are released, the larger the chance of high error on some query.) This is computationally feasible as each sample $G(z)$ is simply a sample from a Gaussian having known covariance with the previous samples drawn.

Unfortunately, it would not be sufficient to independently release $\hat{f}_{1:t}$ at each time $t$, because the amount of noise required would be prohibitive. This leads us to our next tool.

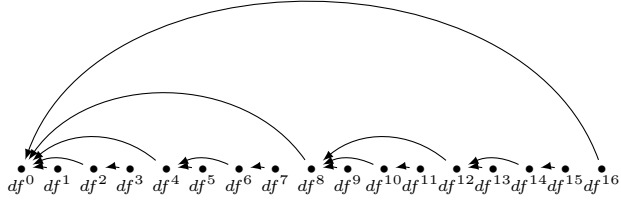

$df^0\ df^1\ df^2\ df^3\ df^4\ df^5\ df^6\ df^7\ df^8\ df^9\ df^{10}df^{11}df^{12}df^{13}df^{14}df^{15}df^{16}$

Figure 1: Picturing the continual observation technique for preserving privacy. Each $df^t$ is a trade (*e.g.* a data point sold to the market). The goal is to release, at each time step $t$, a noisy version of $f^t = \sum_{j=1}^{t} df^j$. To do so, start at $t$ and follow the arrow back to $s(t)$. Take the partial sum of $df^j$ for $j$ from $s(t)$ to $t$ and add some random noise. Trace the next arrow from $s(t)$ to $s(s(t))$ to get another partial sum and add noise to that sum as well. Repeat until 0 is reached, then add together all the noisy partial sums to get the output at time $t$, which will equal $f^t$ plus noise. The key point is that we can re-use many of the noisy partial sums in many different time steps. For instance, the noisy partial sum from 0 to 8 can be re-used when releasing all of $f^9, \ldots, f^{15}$. Meanwhile, each $df^t$ participates in few noisy partial sums (the number of arrows passing above it).

**Tool 2: Continual observation technique.** The idea of this technique, pioneered by [9, 5], is to construct $\hat{f}^t = \sum_{j=0}^{t} df^t$ by adding together noisy partial sums of the form $\hat{f}_{t_1:t_2}$ as constructed in Equation 4. The idea for choosing these partial sums is pictured in Figure 1: For a function $s(t)$ that returns an integer smaller than $t$, we take $\hat{f}^t = \hat{f}^{s(t)+1:t} + \hat{f}^{s(s(t))+1:s(t)} + \cdots + \hat{f}^{0:0}$. Specifically, $s(t)$ is determined by writing $t$ in binary, then flipping the rightmost "one" bit to zero. This is pictured in Figure 1. The intuition behind why this technique helps is twofold. First, the total noise in $\hat{f}^t$ is the sum of noises of its partial sums, and it turns out that there are at most $\lceil \log T \rceil$ terms. Second, the total noise we need to add to protect privacy is governed by how many different partial sums each $df^j$ participates in, and it turns out that this number is also at most $\lceil \log T \rceil$. This allows for much better privacy and accuracy guarantees than naively treating each step independently.

## 3.2 Mechanism and Results

Combining our market template in Mechanism 1 with the above privacy tools, we obtain Mechanism 2. There are some key differences. First, we have a bound $Q$ on the total number of queries. (Each query $x$ returns the instantaneous prices in the market for $x$.) This is because each query reveals information about the participants, so intuitively, allowing too many queries must sacrifice either privacy or accuracy. Fortunately, this bound $Q$ can be an arbitrarily large polynomial in the number of traders without affecting the quality of the results. Second, we have PAC-style guarantees on accuracy: with probability $1 - \gamma$, all price queries return values within $\alpha$ of their true prices. Third, it is no longer straightforward to compute and represent the market prices $\nabla C_x(\hat{f}^t)$ unless $\mathcal{Y}$ is finite. We leave the more general analysis of Mechanism 2 to future work.

Either exactly or approximately, Mechanism 2 inherits the desirable properties of Mechanism 1, such as bounded budget and incentive-compatitibility (that is, participants are incentivized to minimize the risk of the market hypothesis). In addition, we show that it preserves privacy while maintaining accuracy, for an appropriate choice of the price sensitivity $\lambda_C$.

**Theorem 2.** *Consider Mechanism 2, where $\Delta$ is the maximimum trade size (Equation 3) and $d = |\mathcal{Y}|$. Then Mechanism 2 is $(\epsilon, \delta)$ differentially private and, with $T$ traders and $Q$ price queries, has the following accuracy guarantee: with probability $1 - \gamma$, for each query $x$ the returned prices satisfy $\|\nabla C_x(\hat{f}^t) - \nabla C_x(f^t)\|_\infty \leq \alpha$ by setting*

$$\lambda_C = \frac{\alpha\epsilon}{2d\Delta^2 \sqrt{\ln \frac{Qd}{\gamma} \ln \frac{2\log T}{\delta} \log(T)^3}}.$$

If one for example takes $\delta, \gamma = \exp\left[-\text{polylog}(Q, T)\right]$, then except for a superpolynomially low failure probability, Mechanism 2 answers all queries to within accuracy $\alpha$ by setting the price sensitivity to be $\lambda_C = O\left(\alpha\epsilon/\text{polylog}(Q, T)\right)$. We note, however, that this is a somewhat weaker guarantee than is usually desired in the differential privacy literature, where ideally $\delta$ is exponentially small.

---
**Mechanism 2:** Privacy Protected Market

---
Parameters: $\epsilon, \delta$ (privacy), $\alpha, \gamma$ (accuracy), $k$ (kernel), $\Delta$ (trade size 3), $Q$ (#queries), $T$ (#traders)

MARKET announces $\hat{f}^0 = f^0$, sets $r = 0$, sets $C$ with $\lambda_C = \lambda_C(\epsilon, \delta, \alpha, \gamma, \Delta, Q, T)$ (Theorem 2)

**for** $t = 1, 2, \ldots, T$ **do**

    PARTICIPANT $t$ proposes a bet $df^t$

    MARKET updates true position $f^t = f^{t-1} + df^t$

    MARKET instantiates $\hat{f}^{s(t)+1,t}$ as defined in Equation 4

    **while** $r \leq Q$ and some OBSERVER *wishes to make a query* **do**

        OBSERVER $r$ submits pricing query on $x$

        MARKET returns prices $\nabla C_x(\hat{f}^t)$, where $\hat{f}^t = \hat{f}^{s(t)+1:t} + \hat{f}^{s(s(t))+1:s(t)} + \cdots + \hat{f}^{0:0}$

        MARKET sets $r \leftarrow r + 1$

MARKET observes a true sample $(x, y)$

**for** $t = 1, 2, \ldots, T$ **do**

    PARTICIPANT receives payment $f^{t-1}(x, y) - f^t(x, y) - C_x(\hat{f}^{t-1} + df^t) + C_x(\hat{f}^{t-1})$

---

**Computing** $\nabla C_x(\hat{f}^t)$**.** We have already discussed limiting to finite $|\mathcal{Y}|$ in order to efficiently compute the marginal prices $\nabla C_x(\hat{f}^t)$. However, it is still not immediately clear how to compute these prices, and hence how to implement Mechanism 2. Here, we show that the problem can be solved when $C$ comes from an exponential family, so that $C_x(f) = \log \int_{\mathcal{Y}} \exp \left[ f(x, y) \right] dy$. In this case, the marginal prices given by the gradient of $C$ have a nice exponential-weights form, namely the price of shares in $(x, y)$ is $p_x^t(y) = \nabla_y C_x(f^t) = \frac{e^{f(x,y)}}{\sum_{y \in \mathcal{Y}} e^{f(x,y)}}$. Thus evaluating the prices can be done by evaluating $f^t(x, y)$ for each $y \in \mathcal{Y}$.

We also note that the worst-case bound used here could be greatly improved by taking into account the structure of the kernel. For "smooth" cases such as the Gaussian kernel, querying a second point very close to the first one requires very little additional randomness and builds up very little additional error. We gave only a worst-case bound that holds for all kernels.

**Adding a transaction fee.** In the appendix, we discuss the potential need for *transaction fees*. Adding a small $\Theta(\alpha)$ fee suffices to deter *arbitrage* opportunities introduced by noisy pricing.

### Discussion

The main contribution of this work was to bring together several tools to construct a mechanism for incentivized data aggregation with "contest-like" incentive properties, privacy guarantees, and limited downside for the mechanism.

Our proposed mechanisms are also extensions of the prediction market literature. Building upon the work of Abernethy et al. [1] we introduce the following innovations:

- **Conditional markets.** Our framework of Mechanism 1 can be interpreted as a prediction market for conditional predictions $p(y|x)$ rather than a classic market which would elicit the joint distribution $p(x, y)$, or just the marginals. (This is similar to *decision markets* [12, 7], but without out the associated incentive problems.) Naturally then, we couple conditional predictions with *restricted hypothesis* spaces, allowing $\mathcal{F}$ to capture, e.g., a linear relationship between $x$ and $y$.
- **Nonparametric securities.** We also extend to *nonparametric* hypothesis spaces using kernels, following the kernel-based scoring rules of [21].
- **Privacy guarantees.** We provide the first *private prediction market* (to our knowledge), showing that information about individual trades is not revealed. Our approach for preserving privacy also holds in the classic prediction market setting with similar privacy and accuracy guarantees.

Many directions remain for future work. These mechanisms could be made more practical and perhaps even better privacy guarantees derived, especially in nonparametric settings. One could also explore the connections to similar settings, such as when agents have costs for acquiring data.

**Acknoledgements** J. Abernethy acknowledges the generous support of the US National Science Foundation under CAREER Grant IIS-1453304 and Grant IIS-1421391.

## Footnotes

[1]This can easily be extended to a test *set* by taking the average performance over the test set.

[2]Formally, each $G(z)$ is a random variable and, for any finite subset of $\mathcal{Z}$, the corresponding variables are distributed as a multivariate normal with covariance given by $k$.

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
