[Reviews · NeurIPS 2015]

Submitted by Assigned_Reviewer_1

The authors identify a strategy for compensating agents to disclose private and relevant data. The strategy draws on interesting ideas from the machine learning literature, including the use of the stochastic gradient descent algorithm to set the payment for the data. This allows effective compensation of agents while maintaining a limited budget. The authors also include mechanisms for preserving the privacy of agents, and identification of different "profit maximizing" strategies for agent to select given their confidence in their data. It appears that the substantive contribution is

in identifying a mechanism that compensates agents for their data while maintaing a bounded budget.

The usefulness of this family of algorithms hinges on timely, available correct information.

If "true samples" will arise only in the distant future, then section 2.3 is not viable (see below), section 2.1 does not seem to offer much advantage over traditional betting markets (modulo the generalizability to more complex prediction function).

Section 3 does not strike me as addressing a fundamentally serious problem: depending on the type of privacy one is concerned with, there seem to be simpler solutions to keep agent i from learning about agent j: (a) anonymized participation (b) anonymized relations between updates and agents (which agent made which trade is not available to other agents)

The really useful contribution here is a generalization of prediction markets to use sophisticated prediction tools, thus allowing for prediction markets to extend far beyond bets on single binary or scalar variables.

This is cool, but in practice it seems like it would offer little motivation for participation, unless the market happened to run along-side a constant stream of true samples, such that rewards could be evaluated quickly.

(If I won't know what my data is bidding on, how much credit it might get, and when the rewards will come, why should i play?)

Section 2.3 (offering to buy data) seems predicated on knowing the loss function of predictions, which means that the correct answers are known.

In that case, what's even the point of bidding on data?

If the answers are not known, then how does one come up with the marginal gain from a given data point?

Seems to be a pretty fundamental catch-22.

There are a few possible ways out of this, but they all seem to open the system up to being gamed: - Use past true samples to evaluate the worth of new training data.

However, then one can simply offer lots of training data to match known past samples, and reap the rewards by generating fake data to overfit past data. - Use some prediction divergence measure to evaluate the utility of new data (i.e., all data that changes my predictions is good).

This can obviously be gamed by generating crazy data to make predictions veer wildly offtrack.

Thus, it seems that the only way to set up a cost function is in light of *future* true samples, and this seems to be a serious limitation insofar as the future is distant.

It absolutely precludes the kind of "selling" scheme in section 2.3
Summary: The authors propose a family of market algorithms that extend prediction markets by rewarding incremental improvements in prediction from contributed data.

This is a really cool idea, however, the practical applications of this seem very limited because the market requires a known Loss function, so if the object of prediction is not known, then there is no way to set up accurate incentives.

Submitted by Assigned_Reviewer_2

Summary: The authors propose adapting tools from the design and analysis of prediction markets to the problem of learning a good hypothesis when the training data is distributed amongst many parties.

The authors also propose modifications to guarantee differential privacy.

In more detail, the authors propose maintaining a current hypothesis, allowing participants to update this hypothesis (various "betting languages" for doing this are considered), and using a family of convex cost functions (parameterized by a domain element x) to charge/reward participants after the final hypothesis is applied to a test data point.

(The choice of the cost functions is dictated by the loss function.)

Differential privacy is added by adapting the state-of-the-art techniques in "continual observation" models (where one want privacy at each time step, without paying linearly in the number of time steps).

Quality: The stitching together of the various models and techniques is competently done.

The paper feels a bit weak on motivation.

The results have a "here's what we know how to do" flavor to them, as opposed to the more traditional "here's a well-motivated problem" and "here's our solution and why it's better than previous/obvious solutions" narrative.

Clarity: The writing quality is reasonably good.

Originality: None of the tools used are original.

Some of their combinations here appear original.

Significance: I find the results modestly significant.
Summary: The authors propose adapting tools from the design and analysis of prediction markets to the problem of learning a good hypothesis when the training data is distributed amongst many parties.

The authors also propose modifications to guarantee differential privacy.

Author Feedback
Author rebuttal: Thanks to all the reviewers for their time and feedback. We provide some specific responses and clarifications.

Assigned_Reviewer_2:
Thank you for your review and summary which we believe makes good points. Your main criticism appears to be that you found the work poorly motivated. We agree that the paper was somewhat light on this matter, but this was a matter of omission: we do think there is a good deal of motivation for the main goal of the paper, a mechanism for learning in a decentralized fashion with privacy guarantees. See for example existing work in crowdsourcing methods for solving ML problems (e.g. Cai et al, Cummings et al COLT 2015; Abernethy et al EC 2015; Cummings et al ITCS 2015) and with privacy guarantees (e.g. Ghosh-Roth EC 2011, Fleisher-Lyu EC 2012, Ligett-Roth WINE 2012, Ghosh-Ligett-Roth-Schoenebeck EC 2014). Applications include learning on data about health care, shopping habits, web browsing, etc. Our motivation is to design a mechanism offering new features compared to this prior work, such as a bounded budget and a more realistic variety of learning objectives. We certainly agree that we should include more of this motivation in the paper itself.

Assigned_Reviewer_3:
Thank you for your review and summary, which we feel is quite accurate. We offer clarification on privacy (Section 3) and offering to buy data (Section 2.3).

For Section 3, the question of privacy is potentially serious and difficult to guarantee robustly. To traders, competitors inferring their trades can be financially disadvantageous. More importantly, data holders may be very worried about others gaining knowledge of their personal information e.g. medical data. In these types of situations, which arise very commonly in practice, it is hard to see whether (a) and (b) will be sufficient -- for example, an adversary might control many fake/computer traders and observe many trades. Simple anonymization often fails in practice (e.g. Netflix dataset, and countless others since). This motivates differential privacy as formally guaranteeing both quantifiable privacy and accuracy (see e.g. the book by Dwork and Roth for more motivation).

For Section 2.3, there may be some misunderstanding. The reviewer is correct that the net payment to participants depends crucially on the test data / correct answers, which are potentially revealed far into the future (see below). This payment in fact is broken into two pieces, however: the "cost", which can be computed immediately, and the "payoff" (the outcome-dependent value of the purchased securities) which depends on the test data. Moreover, the gradient-like trade for "selling data" can also be computed immediately: there is a single loss function that is selected by the market designer before the market opens (corresponding to the cost function), and selling a data point means executing a trade equal to a gradient descent step of that known loss function using the data point. We do not need any test data to calculate this descent step, or its cost, just the "training data" provided by the participants so far. The reason we don't end up with a catch-22 is that trades are not guaranteed to make the participant money. They should only sell their data if they expect it to improve the prediction on the test data.

On the important question of whether distant-future payoffs are motivating enough, we first note that the reviewer's suggestion to offer a continuous stream of test data and payments is a straightforward extension to our model, and so this proposal may address this objection (and indeed, many ML competitions consist of multiple rounds). Second, we point out cases where distant payoffs work in practice: crowdsourcing or machine-learning contest platforms such as kaggle, and prediction markets such as the Iowa Electronic Market, Betfair, and even SciCast where payments may only come years later. People do participate in these platforms despite this sometimes lengthy temporal separation. Our mechanism would only make participation easier by allowing them to sell data rather than formulate entire algorithms/predictions. The small payments might also add up into a continuous revenue stream with participation in many different markets.